# Oral Polio Vaccine Campaigns May Reduce the Risk of Death from Respiratory Infections

**DOI:** 10.3390/vaccines9101133

**Published:** 2021-10-04

**Authors:** Sebastian Nielsen, Hasan Mahmud Sujan, Christine Stabell Benn, Peter Aaby, Syed Manzoor Ahmed Hanifi

**Affiliations:** 1Bandim Health Project, OPEN (Odense Patient data Explorative Network), Institute of Clinical Research, Studiestræde 6, 1455 Copenhagen, Denmark; cbenn@health.sdu.dk (C.S.B.); p.aaby@bandim.org (P.A.); 2Bandim Health Project, Indepth Network, Apartado 861, Bissau 1004, Guinea-Bissau; 3International Centre for Diarrhoeal Disease Research, Bangladesh (icddr,b), Health System and Population Studies Division, Shaheed Tajuddin Ahmed Sarani, Mohakhali, Dhaka 1212, Bangladesh; hasan.sujan@icddrb.org (H.M.S.); hanifi@icddrb.org (S.M.A.H.); 4Danish Institute of Advanced Science, Odense University Hospital and University of Southern Denmark, Campusvej 55, 5230 Odense, Denmark

**Keywords:** Bangladesh, non-specific effects of vaccines, cause of death, respiratory infections, campaigns, OPV, oral polio vaccine

## Abstract

Oral polio vaccine (OPV) campaigns, but not other campaigns, have been associated with major reductions in child mortality. Studies have shown that OPV reduces the risk of respiratory infections. We analysed the causes of death at 0–2 years of age in Chakaria, a health and demographic surveillance Systems in Bangladesh, in the period 2012–2019 where 13 national campaigns with combinations of OPV (*n* = 4), vitamin A supplementation (*n* = 9), measles vaccine (MV) (*n* = 2), and albendazole (*n* = 2) were implemented. OPV-only campaigns reduced overall mortality by 30% (95% confidence interval: −10–56%). Deaths from respiratory infections were reduced by 62% (20–82%, *p* = 0.01) in the post-neonatal period (1–35 months), whereas there was as slight increase of 19% (−37–127%, *p* = 0.54) for deaths from other causes. There was no benefit of other types of campaigns. Hence, the hypothesis that OPV may have beneficial non-specific effects, protecting particularly against respiratory infections, was confirmed.

## 1. Introduction

Several studies from Bangladesh, Burkina Faso, Ghana and Guinea-Bissau have found that campaigns with oral polio vaccine (C-OPV) are associated with marked reductions in the child mortality rate [1,2,3,4,5]. There has been no study in low-income countries of whether this beneficial effect of C-OPV is general or specific to certain infections.

When oral polio vaccine (OPV) was first introduced in the 1960s, it was suggested that OPV particularly reduced diarrhoea morbidity and mortality in Latin America [6] and in studies in the Soviet Union showed that OPV in particular had an effect on respiratory infections [7,8]. Randomised control trials (RCTs) have compared OPV with inactivated polio vaccine (IPV) in Finland and Bangladesh and found that OPV protected against upper respiratory infections and diarrhoea, respectively, but only these outcomes were examined in the studies [9,10]. There has been no specific examination of which infections or potential causes of deaths are most affected by C-OPVs. 

In recent light of the COVID-19 pandemic, a better understanding of OPV is much needed to harness the potential beneficial non-specific effects of this vaccine [11,12,13].

We used data from Chakaria, a Health and Demographic Surveillance Systems (HDSS) in Bangladesh, to examine this question. From 2012 to 2019, the HDSS conducted verbal autopsies of all child deaths. We previously reported that C-OPVs reduced under-three mortality by 31% (95% CI: 10–48%) between 2004 and 2019 [1]. In connection with a change in data collection platform in 2012, the HDSS started to collect verbal autopsies. In the 2004–2011 period there were 17 C-OPVs and the mortality rate in the community declined 34% (4–54%) after C-OPV-only. In the subsequent period from 2012–2019, there were only four C-OPVs between January 2012 and January 2014 and the reduction in mortality was 32% (−4–56%) after C-OPV-only [1]. 

## 2. Materials and Methods

### 2.1. Setting

The Chakaria HDSS, located on the southeastern coast of the Bay of Bengal, was initiated in 1999 and is one of the field sites of the International Centre for Diarrhoeal Disease Research in Bangladesh (icddr,b). Data collection was interrupted between 2001 and 2003 and the framework for data collection differed over time because the HDSS in the period 2004–2011 followed a random selection of households, whereas a random sample of villages was followed after 2012. From 2012–2019, the HDSS covered 49 villages among all 183 villages in the study area with 86,000 residents living in 17,000 households. All households from the chosen villages were followed with quarterly home visits where information on outcomes and socio-economic variables is collected. In the second period children were also registered during pregnancy and verbal autopsies were introduced. At the home visits, information was collected on all individuals alive at the time of the visit. For the present analysis only the second period is included.

### 2.2. Exposure and Outcomes

As described elsewhere [1], information on campaigns was obtained from the Chakaria HDSS and from the local World Health Organisation (WHO) office in Bangladesh. We used this information unless otherwise stated [1]. We had also data on OPV campaigns from the WHO international office in Geneva that provided a database of all OPV campaigns conducted globally. Compared to the data from Bangladesh, the global WHO data were less accurate and in some cases wrong, in the sense that there was certainty locally that some campaigns registered in the international data had not been carried out. Information on other campaigns providing vitamin A supplementation (VAS) and measles vaccine (MV) was obtained from the Chakaria HDSS. 

As in previous analyses [1,4,5], we focused on OPV-only campaigns, but we also conducted an analysis for any-OPV, i.e., including also campaigns where OPV was given at the same time as other interventions. Information on survival came from the HDSS. 

### 2.3. Verbal Autopsies

Verbal autopsies were collected by a team of trained surveillance workers at the Chakaria HDSS. Information regarding circumstance of death, signs and symptoms leading to death were obtained from close family members. Cause of death was categorised using a verbal autopsy algorithm [14,15]. In the present analysis, all deaths before 29 days of age were categorised as “Neonatal deaths”; these were excluded from all the analyses. “Respiratory infections” were defined as: acute respiratory infection, pneunomia, pertussis, measles and asthma. “Other causes” were all other categories apart from “Accidents” and deaths with “Missing cause” (see Appendix A).

### 2.4. Statistical Methods

The analysis of the effect of campaigns on mortality has been described previously [1]. Briefly, using Cox proportional hazards models, we calculated hazard ratios (HR) comparing mortality for children under 3 years of age “after C-OPV” vs. “before C-OPV” to assess the effect of receiving a C-OPV. The proportional hazards assumption was violated for the OPV + MV-campaign covariate for the analysis of respiratory deaths; this problem was solved by stratifying analysis by before and after 1 January 2016. (See Appendix A).

National immunisation campaigns are essentially a natural experiment since they allocate children to participate in campaigns based on their birthday and thereby eliminate many common causes of bias. For these reasons, most potential confounders, such as socio-economic status (SES) or education, would not confound the estimated association between campaign participation and risk of mortality.

We present results from two Cox models. In the first Cox model we adjusted for age and included an interaction term (c.year#i.agegroup) for the continuous covariate of years since 2012 (year) and age groups: 0–5, 6–11, 12–23, and 24–35 months of age (agegroup); using finer timer intervals did not change the results (data not shown). As in previous analyses [1], we present the HR results for infants and older children, and the two groups combined.

The second model, representing the main analysis, included adjustment in the multivariable analysis for all campaigns, including campaigns with OPV-only (C-OPV-only, *n* = 2), campaigns with OPV and vitamin A supplementation (C-OPV + VAS, *n* = 1), campaigns with OPV and measles vaccine (C-OPV + MV, *n* = 1), campaigns with VAS-only (C-VAS-only, *n* = 8), and campaigns with MV-only (C-MV-only, *n* = 1). Since albendazole was always given with VAS, the separate effect of albendazole was not analysed. In the second model for the primary analysis, the proportional hazards assumption was violated for C-OPV+MV (*p* = 0.004), we therefore allowed the baseline hazards function to be different for the two calendar periods, 2012–2014 and 2015–2019. This solved the problem (*p* = 0.05), but also changed the estimate for C-OPV-only from 0.49 (0.24–1.01) to 0.38 (0.18–0.80).

For the period 2012–2019, we analysed whether the campaigns were associated with similar effects for different causes of deaths. In the 2012–2019 period, there were 496 deaths for children under three years of age. Of these deaths, 169 occurred in the neonatal period; among the 327 post-neonatal deaths, 72 were caused by accidents, 134 respiratory infections (including 4 deaths from measles and 2 from pertussis), 103 other causes, and 18 had no cause assigned. As seen in Appendix A, respiratory deaths were mainly in the infant period (82%) as were other infectious diseases (67%), deaths with “no cause assigned” were 72% infant, but only 10% of accidental deaths occurred among infants. The “no cause assigned” group was too small to be analysed. C-OPV-only was associated with a non-significant reduction in accidents deaths but was not further analysed since campaigns presumably have no effect on accidental death. Since neonatal deaths are likely to have a different aetiology and only 1.0% of follow-up time in the neonatal period was after C-OPV (Appendix A); we disregarded neonatal deaths in the analysis of the effect of campaigns on cause of death.

The level of statistical significance was defined as *p* < 0.05.

## 3. Results

During 2012–2019 a total of 496 children died and were included in the previously reported overall mortality analysis [1], among these 27% (*n* = 134) were caused by respiratory infections, 21% (*n* = 103) by other causes, 15% (*n* = 72) by accidents, 34% (*n* = 169) were neonatal deaths and 4% (*n* = 18) were indeterminate or had missing cause.

As seen in Table 1, C-OPV-only was associated with HR of 0.38 (0.18–0.80, *p* = 0.01) for death from respiratory infections whereas the HR was 1.19 (0.63–2.27, *p* = 0.54) for deaths from other causes (*p* = 0.023, test of same effect of C-OPV-only on respiratory infections and other causes). There was no difference by sex (data not shown). The effects in infancy and childhood were similar for both respiratory infections and other causes (Appendix A). If all OPV campaigns were combined to “any-OPV campaign”, the HR for death from respiratory infections was 0.59 (0.31–1.15, *p* = 0.12) (Appendix A).

Among “other causes”, meningitis (*n* = 17), diarrhoea (*n* = 15), HIV/AIDS-related death (*n* = 13), and severe malnutrition (*n* = 10) were the largest subgroups. When diarrhoea and HIV were tested separately, the HRs were 2.57 (0.84–7.86, *p* = 0.10) and 3.57 (0.91–14.0, *p* = 0.07), respectively. The main multivariable model did not converge for meningitis, severe malnutrion and missing cause of death; the HRs for these in the first model were 0.77 (0.23–2.57, *p* = 0.67), 0.44 (0.05–4.25, *p* = 0.48) and 2.07 (0.70–6.12, *p* = 0.19), respectively. The estimate for accidents was 0.40 (0.14–1.11, *p* = 0.08) in the main multivariable model.

## 4. Discussion

OPV-only campaigns have been associated with reduced all-cause mortality. In the present study, OPV-only campaigns significantly reduced the risk of dying of respiratory infections. There was no indication of effects on other infectious causes of death, but numbers were limited. Hence, the study corroborated the hypothesis that the nonspecific beneficial effect of OPV may be strongest for respiratory infections.

The effect of C-OPV-only on cause of death has not been tested before. However, other studies have suggested a strong effect of routine OPV on respiratory infections. 

Denmark used live OPV in three doses at 2, 3 and 4 years of age until 2001. Having received OPV rather than diphtheria, tetanus, acellular pertussis, inactivated poliovirus, haemophilus B conjugate vaccine (DTaP–IPV–Hib) as the most recent vaccination was associated with 27% (13–39%) lower risk of being hospitalised with lower respiratory infection in the third year of life, whereas the risk reduction was 7% (−10–21%) for upper respiratory infections, 11% (−18–33%) for gastrointestinal infections, and 6% (−20–26%) for other infections [16]. 

In a natural experiment in Guinea-Bissau, when diphtheria, tetanus and pertussis vaccine (DTP) were missing for several months, it was possible to compare the case fatality ratio (CFR) of children who had received OPV-only or OPV+DTP. OPV-only-recipients compared with OPV+DTP-recipients had 70% (19–89%) lower CFR; the reduction was 68% (−125–95%) for pneumonia, 84% (−14–98%) for presumptive malaria, 57% (−200–94%) for diarrhoea, and 48% (−606–96%) for other infections [17]. 

RCTs of other live vaccines, Bacillus Calmette-Guérin (BCG) and measles vaccine (MV), have also been linked to lower risk of dying of non-targeted infections in population-based studies. In three RCTs among low weight children, who normally did not receive BCG at birth in Guinea-Bissau, BCG-at-birth vs. no BCG was associated by 38% (17–54%) lower neonatal mortality. The reduction was for infectious diseases, with no effect for non-infectious diseases [18]. To obtain better diagnoses for the infectious causes, we used the hospitalisation records for all participants in the three trials [19]. There was little effect on the risk of hospitalisation, but BCG vaccinated children had 54% (2–78%) lower CFR for neonatal sepsis. BCG was associated with fewer admissions for pneumonia in these neonatal cohorts, but no difference in diarrhoea and malaria admissions. In an RCT of BCG-at-birth versus BCG-at-6 weeks (time of DTP vaccination) in Uganda, the early BCG-vaccinated children had 29% (5–47%) less physician-diagnosed non-tuberculosis infectious disease episodes (*n* = 98 events) than the controls (*n* = 129 events) between birth and 6 weeks of age; the difference of 31 events was mainly due to 23 fewer episodes of upper and lower respiratory tract infections among the BCG-vaccinated children [20].

In an RCT of early MV vs. no-MV at 4.5 months of age, we could examine how early MV affected the risk of hospital admissions between 4.5 and 9 months of age, when everybody received MV. MV reduced the risk of admissions with measles by 100% (56–100%) and respiratory infections by 63% (11–84%) and malaria by 38% (−26–68%), but there was no effect for diarrhoea and other infections [21]. 

In conclusion, OPV-only campaigns seemed to have the strongest effect for respiratory infections. However, it should be noted that this was for OPV-only campaigns and not for any kind of campaign combining OPV with other interventions. There was little evidence for a special effect on diarrhoea [6]. Other live vaccines, BCG and MV, have also been observed to have the strongest effect for respiratory infections and sepsis and little effect for diarrhoea. The evidence for an effect on malaria is less consistent. 

## Figures and Tables

**Table 1 vaccines-09-01133-t001:** Mortality rates (per 100 person-years) and hazard ratios (HR) for after-campaign vs. before-campaign for children only eligible for the oral polio vaccine (OPV) by cause of death. Analysis from 29 days to 35 months of age among children within the Chakaria HDSS from 2012 to 2019.

Respiratory Infections				
Campaign	After-Campaign Mortality Rates per 100 Years (Deaths/Person Years)	Before-Campaign Mortality Rates per 100 Years (Deaths/Person Years)	HR (After-/Before-Campaign) (95% CI) ^1^	Main Model HR (After-/Before- Campaign) (95% CI) ^2^
Campaign-OPV-only	0.12 (15/12,520)	0.42 (119/28,233)	0.48 (0.26–0.91) *	0.38 (0.18–0.80) *
Campaign-OPV + VAS	0.14 (7/5043)	0.36 (127/35,711)	0.71 (0.33–1.55)	1.28 (0.58–2.81)
Campaign-OPV + MV	0.13 (6/4771)	0.36 (128/35,983)	1.73 (0.50–5.97)	2.70 (0.66–11.0)
Campaign-VAS-only	0.14 (32/22,968)	0.57 (102/17,785)	1.14 (0.64–2.04)	1.04 (0.55–1.95)
Campaign-MV-only	0.06 (3/4830)	0.36 (131/35,923)	0.92 (0.21–3.94)	0.87 (0.19–4.05)
**Other causes**				
Campaign	After-campaign mortality rates per 100 years (deaths/person years)	Before-campaign mortality rates per 100 years (deaths/person years)	HR (After/Before-campaign) (95% CI) ^1^	Main model HR (After/Before- campaign) (95% CI) ^2^
Campaign-OPV-only	0.19 (24/12,520)	0.28 (79/28,233)	0.95 (0.49–1.84)	1.19 (0.63–2.27)
Campaign-OPV + VAS	0.18 (9/5043)	0.26 (94/35,711)	0.64 (0.31–1.33)	0.61 (0.26–1.42)
Campaign-OPV + MV	0.13 (6/4771)	0.27 (97/35,983)	0.86 (0.26–2.80)	0.71 (0.21–2.43)
Campaign-VAS-only	0.18 (42/22,968)	0.34 (61/17,785)	1.45 (0.78–2.68)	1.30 (0.68–2.49)
Campaign-MV-only	0.12 (6/4830)	0.27 (97/35,923)	1.60 (0.63–4.08)	1.51 (0.56–4.04)

^1^ Adjusting for age (underlying time) and year * age group. ^2^ Main multivariable model: adjusting for age (underlying time), OPV, OPV + VAS, OPV + MV, VAS, MV and year * age group. * *p* < 0.05. VAS = vitamin A supplementation, MV = measles vaccine.

## Data Availability

The datasets used for these analyses are not publicly available, but access can be requested to the authors.

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
