# Peer review of "Oral Polio Vaccine Campaigns May Reduce the Risk of Death from Respiratory Infections"

_vaccines, 2021, doi:10.3390/vaccines9101133_

Round 1

Reviewer 1 Report

The MS deals with an important issue: (innate) immunity promoted by vaccine campaigns unexpectedly associated with the prevention of disease/death from infections not addressed by the employed vaccine.

Authors should better introduce the theme that is little known to many potential readers. To this end, three general references seem adequate; additional references referring to Africa and other countries are already present in the MS.

1: Chumakov MP, Voroshilova MK, Antsupova AS, BoÄ­ko VM, Blinova MI, PriÄ­miagi LS, Rodin VI, SeÄ­bil' VB, Siniak KM, Smorodintsev AA, et al. Zhivye énterovirusnye vaktsiny dlia ékstrennoÄ­ nespetsificheskoÄ­ profilaktiki massovykh respiratornykh zabolevaniÄ­ vo vremia osenne-zimnikh épidemiÄ­ grippa i ostrykh respiratornykh zabolevaniÄ­ [Live enteroviral vaccines for the emergency nonspecific prevention of mass respiratory diseases during fall-winter epidemics of influenza and acute respiratory diseases]. Zh Mikrobiol Epidemiol Immunobiol. 1992;(11-12):37-40. Russian. PMID: 1338742.

2: Voroshilova MK. Potential use of nonpathogenic enteroviruses for control of human disease. Prog Med Virol. 1989;36:191-202. PMID: 2555836.

3: Chumakov K, Avidan MS, Benn CS, Bertozzi SM, Blatt L, Chang AY, Jamison DT, Khader SA, Kottilil S, Netea MG, Sparrow A, Gallo RC. Old vaccines for new infections: Exploiting innate immunity to control COVID-19 and prevent future pandemics. Proc Natl Acad Sci U S A. 2021 May 25;118(21):e2101718118. doi: 10.1073/pnas.2101718118.

The referral system for southern Bangladesh needs to be better defined:

Chakaria Health and Demographic Surveillance System (CHDSS), located on the south-eastern coast of the Bay of Bengal, was established in 1999 and is one of the field sites of International Centre for Diarrhoeal Disease Research, Bangladesh (ICDDRB). The surveillance covers 118 315 residents living in 19 847 households. Data on socio-demographic and health indicators including birth, death, migration, marriage, maternal health, education and employment are recorded through quarterly household visits. The primary objective of CHDSS is to monitor the changes in socio-demographic indicators, inequalities in health and impact of public health interventions. A demographic change was accompanied by a shift from traditional to modern society during the past decade, but inequality in health still persists. The findings from the surveillance are shared regularly among the local and global communities. Data are also available upon request to ICDDRB and INDEPTH for use by researchers and policy makers.

The methods of quarterly visits to families in villages and the Verbal Autopsy method need also to be introduced:

Verbal autopsy (VA) is a method of determining individuals’ causes of death and cause-specific mortality fractions in populations without a complete vital registration system. Verbal autopsies consist of a trained interviewer using a questionnaire to collect information about the signs, symptoms, and demographic characteristics of a recently deceased person from an individual familiar with the deceased. A standard VA instrument paired with easy-to-implement and effective analytic methods can help bridge significant gaps in information about causes of death, particularly in resource-poor settings.

The local and world WHO databases that have been consulted need to be referred to in detail (e.g., inserting the appropriate URL).

Statistical Methods:

please re-group the different analyses made in a way understandable to the reader. Possibly, insert a Table. All through the MS, the resultant P values need to be expressed.

Table 1, two sections:

  • Respiratory infections (need to be defined in the Methods section)
  • Infections other than respiratory

Discussion needs to be more focalized on the indications of the study that are statistically significant, not all indications/suggestions of comparative analyses.

Reviewer 2 Report

The paper provides an analysis of the effect of immunization campaigns with oral anti-poliomyelitis vaccine (OPV), measles vaccine, and vitamin A supplementation on children mortality using data from health and demographic surveillance in Bangladesh. The results demonstrate a strong effect of only the OPV campaigns on the reduction of childhood mortality, in particular from respiratory infections. The paper also provides a useful concise summary of other studies analyzing the protective effect of live vaccinations against non-targeted infections. The data support the growing consensus on the importance of non-specific immunity and provide guidance on possible first-line interventions against rapidly spreading novel infectious diseases.  The intrinsic randomization of children population for the vaccination campaigns based on birth date only, and the thorough critical analysis of the morbidity and mortality data from this study and other sources are important strong points of this work.

The paper is focused and well written, I have only minor suggestions for improvement of the manuscript:

Since albendazole was never administered on its own, the emphasis on its analysis in the Abstract is misleading. Albendazole/vitamin A should be described as one intervention. Moreover, in the Methods section (first paragraph, page 3), it is not albendazole but mebendazole that was given with vitamin A. This should be clarified.

In the discussion, DTaP-IPV-Hib and DTP should be spelled out.

Round 2

Reviewer 1 Report

Upon modest re-editing and corrections, the paper will be ready for publication.

At this time, three points need attention:

1) better definition of "other infections"

2) clear definition o significant p values (traditionally, <0.05, <0.01, <0.001).

0.01<p<0.05 is not clear

VAs - needs to be consistent through the text

Vitamin A supplementation, needs to be distinguished from VAs - I'm proposing VitAS.

For convenience, the above points have been highlighted in the text.
